# NODE-SAT: Temporal Graph Learning with Neural ODE-Guided Self-Attention

## Abstract

We propose NODE-SAT, a novel temporal graph learning model that integrates Neural Ordinary Differential Equations (NODEs) with self-attention mechanisms. NODE-SAT's design requires only historical 1-hop neighbors as input and comprises three key components: a temporal link processing module utilizing NODE-guided self-attention layers to capture temporal link information, a node representation module summarizing neighbor information, and a prediction layer. Extensive experiments across thirteen temporal link prediction datasets demonstrate that NODE-SAT achieves state-of-the-art performance on most datasets with significantly faster convergence. The model demonstrates high accuracy, rapid convergence, robustness across varying dataset complexities, and strong generalization capabilities in both transductive and inductive settings in temporal link prediction. These findings highlight NODE-SAT's effectiveness in capturing node correlations and temporal link dynamics.

## 1 Introduction

Temporal graphs have emerged as a powerful tool for modeling complex, evolving systems across various domains Kazemi et al. (2020); Yu et al. (2017); Bui et al. (2022). These time-varying structures represent entities as nodes and their interactions as timestamped links, capturing the chronological evolution of relationships in diverse scenarios. In social networks, temporal graphs are highly effective at analyzing user interactions and predicting future connections Kumar et al. (2019); Song et al. (2019). Within e-commerce platforms, they play a crucial role in modeling user-item interactions and recommending products Li et al. (2020); Fan et al. (2021); Yu et al. (2022); Zhang et al. (2022b). In the field of transportation, temporal graphs prove invaluable for analyzing traffic patterns and optimizing routes Yu et al. (2017); Wu et al. (2019); Guo et al. (2019). Furthermore, they have shown great promise in modeling and predicting the behavior of complex physical systems Huang et al. (2020b); Sanchez-Gonzalez et al. (2020). Representation learning on temporal graphs can be categorized into continuous-time and discrete-time Huang et al. (2020a). This study focuses on continuous-time representation learning. Unlike discrete-time methods that aggregate interactions into fixed intervals, continuous-time models maintain the exact timing of events, allowing for a more nuanced understanding of the graph's evolutionary patterns.

Conventional approaches to temporal graph learning typically integrate Recurrent Neural Networks (RNNs), temporal attention mechanisms, and Graph Neural Networks (GNNs) to model both temporal information and structural relationships Trivedi et al. (2019); Xu et al. (2020). However, a recent study introducing GraphMixer Cong et al. (2023) challenges this complexity with a simple design. GraphMixer utilizes only 1-hop neighbor information, link features, and node features, processing temporal graph data through an MLP-Mixer Tolstikhin et al. (2021) architecture. Despite its simplicity, GraphMixer achieves performance comparable to or surpassing more complex models across various temporal graph learning tasks. The success of GraphMixer raises important questions about the necessity of complex architectures in temporal graph analysis. Its effectiveness lies in the efficient integration of spatial and temporal information. This finding suggests that simplicity can match or exceed the performance of more sophisticated approaches in capturing temporal graph learning.

While GraphMixer demonstrates effectiveness in general temporal graph learning tasks, specific domains such as traffic prediction require tailored approaches to address their unique challenges.

Traffic prediction, a critical component of intelligent transportation systems, demands models that can capture complex spatio-temporal dependencies and evolving patterns in urban mobility. In this context, Neural Ordinary Differential Equations (NODEs) Chen et al. (2018) have emerged as a promising framework Fang et al. (2021); Choi et al. (2022). NODEs, which model continuous-time dynamics, offer a mathematically approach to representing the fluid nature of traffic flows and their temporal evolution.

Drawing inspiration from GraphMixer Zhang et al. (2022a) and NODEs Chen et al. (2018); Poli et al. (2019), we propose a novel method to model temporal graph dynamics. Our method leverages continuum-depth models to capture the intricate evolution of temporal graphs, utilizing NODE frameworks to model the continuous-time dynamics of graph structures. This method enables us to learn differential equations that describe the temporal evolution of link features and node features, providing a more nuanced and continuous representation of graph dynamics. By integrating these concepts, our method offers a more detailed and smooth perspective on how graphs evolve over time, capturing complex interactions and changes in graph structures.

Our method employs an ODE layer to integrate hidden representations across the temporal graph, yielding continuous-time temporal embeddings for each node. Our model's ability to identify and analyze complex temporal patterns is enhanced through the incorporation of a self-attention mechanism Vaswani et al. (2017). The attention layer is applied to the representations generated by the ODE solver. By combining ODE-based continuous-time modeling with self-attention, our method effectively captures node features and richer temporal link information. This integration allows for a more comprehensive understanding of temporal dynamics in graph data. To encourage future research, we have made NODE-SAT available at https://anonymous.4open.science/r/NODE-SAT-6F12. Our key contributions can be summarized as follows:

1. **Novel Temporal Graph Neural Network Architecture:** We propose a new architecture that combines continuous-time modeling with graph neural networks, specifically designed to capture the dynamic nature of temporal graphs.

2. **Continuous-Time Temporal Embeddings:** Our method generates continuous-time temporal embeddings for each node by integrating hidden representations using an ODE solver, capturing the evolving nature of temporal link and graph structure.

3. **Robust Framework for Temporal Graph Analysis:** By combining NODEs with self-attention, we provide a robust framework for modeling and predicting links in temporal graphs, offering a continuous-time perspective that can potentially capture subtle temporal dynamics often overlooked by discrete-time models.

## 2 PRELIMINARY AND EXISTING WORKS

### 2.1 TEMPORAL LINK PREDICTION

Temporal link prediction in temporal graphs involves analyzing the evolution of a network $G(V, E, T)$ over time, where $V$ is the set of nodes, $E$ is the set of edges, and $T$ represents the time steps $\{t_1, t_2, ..., t_n\}$. Given the observed graph states $G_t$ for $t \in \{t_1, ..., t_{n-1}\}$, our goal is to predict the probability $P(e_{ij}|G_{t_n})$ of a link $e_{ij}$ forming between nodes $v_i$ and $v_j$ at the future time $t_n$.

Figure 1 illustrates the temporal link prediction process. The solid lines represent known connections between nodes at different time steps ($t_1$ to $t_4$). The dashed red lines indicate potential future connections at time $t_4$ that the model aims to predict.

## 3 RELATED WORKS

### 3.1 TEMPROAL GRAPH LEARNING

Temporal graph learning methods model network evolution over time, with various approaches addressing this challenge. JODIE Kumar et al. (2019) uses recurrent neural networks (RNNs) to update node representations based on past interactions, while DySAT Sankar et al. (2020) employs

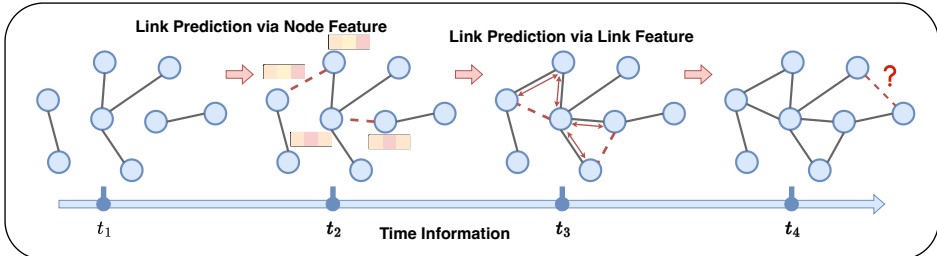

Figure 1: Temporal link prediction with node feature and Link feature along with time information. Specifically, the model is trained to predict the ground-truth links via node and link features at $t_2$ and $t_3$. Note that the examples on $t_2$ and $t_3$ demonstrate how two types of features contribute to the link prediction process. The ultimate goal is to predict the unobserved links at the current time $t_4$.

self-attention on graph snapshots to capture structural changes. TGAT Xu et al. (2020) combines spatial and temporal information by augmenting node features with time encoding. TGN Rossi et al. (2020) merges RNNs with graph attention for joint temporal and spatial modeling. DyGFormer Yu et al. (2023) utilizes Transformers Vaswani et al. (2017) to extract long-term temporal information, and GraphMixer Cong et al. (2023) offers a simple method with a combination of link and node encoders and MLP-Mixer to summarize the information. These methods aim to effectively capture both spatial and temporal information in temporal graphs, enabling applications like link prediction and node classification.

## 3.2 NEURAL ODE

Complex dynamic systems can be modeled using a set of nonlinear first-order ordinary differential equations (ODEs) Strogatz (2018); Guckenheimer & Holmes (2013); Kuznetsov (2013). These ODEs describe the temporal evolution in continuous time $t \in \mathbb{R}$. Let $\mathbf{x}_i(t) \in \mathbb{R}^k$ represent the state vector of the $i$-th variable at time $t$, and $\mathcal{F}$ denote the ODE function governing the system's dynamics. Given the initial conditions $\mathbf{x}_1(0), \mathbf{x}_2(0), \ldots, \mathbf{x}_M(0)$ and the function $\mathcal{F}$, the system's evolution can be solved using numerical ODE solvers such as the Runge-Kutta method Press et al. (2007). The solution for any variable $i$ at an arbitrary time $\tau$ can be expressed as:

$$\mathbf{x}_i(\tau) = \mathbf{x}_i(0) + \int_0^\tau \mathcal{F}(\mathbf{x}_1(t), \mathbf{x}_2(t), \ldots, \mathbf{x}_M(t), t) \, dt \tag{1}$$

This formulation allows for the evaluation of the system's state at any desired time point, providing a continuous representation of the dynamic process.

Neural Ordinary Differential Equation (NODE) Chen et al. (2018) is a continuous-depth deep neural network model. It represents the derivative of the hidden state with a neural network:

$$\frac{d\mathbf{h}(t)}{dt} = \Phi(\mathbf{h}(t), \boldsymbol{\theta}, t) \tag{2}$$

where $\mathbf{h}(t)$ denotes the hidden state of a dynamic system at time $t$, $\Phi$ is a function parameterized by a neural network describing the derivative of the hidden state with respect to time, and $\boldsymbol{\theta}$ represents the parameters in the neural network. The output of a NODE framework is calculated using an ODE solver with an initial value:

$$\mathbf{h}(\tau_1) = \mathbf{h}(\tau_0) + \int_{\tau_0}^{\tau_1} \Phi(\mathbf{h}(t), t, \boldsymbol{\theta}) \, dt \tag{3}$$

where $\tau_0$ is the initial time point, $\tau_1$ is the output time point, and $\mathbf{h}(\tau_1)$ and $\mathbf{h}(\tau_0)$ represent the hidden state at $\tau_1$ and $\tau_0$, respectively. Thus, NODE can output the hidden state of a dynamic system at any time point and deal with continuous-time data, which is extremely useful in modeling continuous-time dynamic systems.

Traditionally, the ODE function $\mathcal{F}$ is usually hand-made based on domain knowledge, such as robot motion control and fluid dynamics Murray (2017); Huang et al. (2023). This approach is challenging without an extensive understanding of the underlying principles. NODEs parameterizing $\mathcal{F}$ with a neural network and learning it in a data-driven way. This approach combines neural networks with ODEs, showing strong results in many different fields.

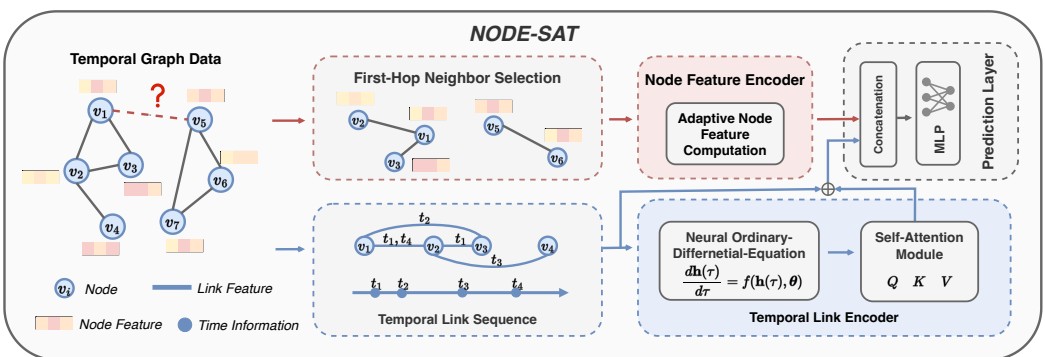

Figure 2: Framework of proposed model.

## 4 METHOD

Our model learns continuous-time representations of nodes in temporal graphs through three key components. The link processing module employs a neural ODE to guide a self-attention layer, capturing complex temporal link information. The node processing module generates node representations that capture structural information from the graph. A prediction layer utilizes these learned representations to predict links at future time points. Figure 2 is the framework of our method.

### 4.1 NODE-SAT

In this section, we present the framework of NODE-SAT, detailing its architecture and key components. For each node $v_i$ in the graph, we represent its temporal link information as a chronologically ordered sequence:

$$T(v_i) = (t_1, \mathbf{x}_{i,j_1}(t_1)) \oplus \cdots \oplus (t_n, \mathbf{x}_{i,j_n}(t_n)) \tag{4}$$

where $t_k$ represents timestamps with $t_1 < \cdots < t_n$, $\mathbf{x}_{i,j}(t)$ denotes the link features between nodes $v_i$ and $v_j$ at time $t$, and $\oplus$ is the concatenation operator. To balance computational efficiency with the recency of information, we retain only the $K$ most recent temporal link entries, where $K$ is a hyperparameter.

To enhance the model's ability to capture temporal patterns, we apply a learnable time encoder Xu et al. (2020). For each timestamp $t_k$, we compute a time encoding vector $\phi(t_k)$ using a learnable function:

$$\phi(t_k) = cos(t_k w + b) \tag{5}$$

where $w$ and $b$ are learnable parameters. This time embedding $\phi(t_k)$ is then concatenated with the link features:

$$\tilde{T}(v_i) = (\mathbf{x}_{i,j_1}(t_1); \phi(t_1)) \oplus \cdots \oplus (\mathbf{x}_{i,j_n}(t_n); \phi(t_n)) \tag{6}$$

To ensure uniform input dimensions, we apply zero-padding to standardize the length of all temporal vectors $\tilde{T}(v_i)$. Following the padding operation, we employ a MLP to summarize the information contained in these vectors:

$$\mathbf{h}_i(T) = \text{MLP}(\tilde{T}(v_i)) \tag{7}$$

where $h_i(T)$ is the embedding of $v_i$ that contains the temporal link information from $t_1$ to $t_n$. The final temporal link embedding is generated through a process that incorporates NODEs, allowing us to model the continuous-time evolution of temporal information of each node:

$$\mathbf{h}_i(T)_\tau = \mathbf{h}_i(T)_{\tau_0} + \int_{\tau_0}^{\tau} f(\mathbf{h}_i(T), \theta)\, dT \tag{8}$$

where $\mathbf{h}_i(T)_\tau$ is the final temporal link embedding for node $v_i$ at future time $\tau$, $\tau_0$ is the current time, $f$ is a neural network, $\theta$ representing parameter of the neural network. we consider the time parameter of the ODE, $\tau$, as a hyperparameter. The integration of NODEs into our model architecture provides a powerful mechanism for controlling the evolution of representations. This method allows us to model the continuous-time dynamics of the graph structure more accurately. We can

estimate temporal link representations at any arbitrary future time point $\tau$. This capability enhances the model's flexibility and generalizability. In our model, higher $\tau$ values allow the temporal link embeddings to evolve over a longer time span, capturing more extended temporal dynamics, while lower $\tau$ values focus on shorter-term changes.

After the output from the Neural ODE, we introduce a self-attention layer to further process and integrate the temporal link information. This step allows the model to capture complex interrelationships between different time points, generating a richer final representation. The self-attention layer Vaswani et al. (2017) is computed as follows:

$$\text{Attention}(Q, K, V) = \text{Softmax}\left(\frac{QK^\top}{\sqrt{d_k}}\right) V \tag{9}$$

where $Q$, $K$, and $V$ are the Query, Key, and Value matrices respectively, and $d_k$ is the dimension of the key vectors. In our case, $Q$, $K$, and $V$ all originate from the same input $\mathbf{h}(v_i)$. The final temporal link representation is generated by combining the original temporal link representation with the output of the attention layer:

$$\mathbf{h}(v_i) = \mathbf{h}_i(T) + \text{Attention}(\mathbf{h}_i(T)_\tau, \mathbf{h}_i(T)_\tau, \mathbf{h}_i(T)_\tau) \tag{10}$$

Here, $\mathbf{h}_i(T)$ is original temporal link representation, $\mathbf{h}_i(T)_\tau$ is the temporal link representation after being processed by NODE, and $\text{Attention}(\mathbf{h}_i(T)_\tau, \mathbf{h}_i(T)_\tau, \mathbf{h}_i(T)_\tau)$ is the output of the self-attention. In this way, our model is able to capture dynamic changes in the continuous time domain using NODEs, learn interdependencies between different time points through the self-attention mechanism, and combine original link information with temporally evolved information to generate a more comprehensive representation for each node. The final representation $\mathbf{h}(v_i)$ contains much richer temporal link information, considering both the continuous time evolution and the relationships between discrete time points.

After processing the temporal link information, we now shift our focus to node features in temporal graphs. Following the studys of GraphMixer Zhang et al. (2022a) and DyGformer Yu et al. (2023), we also adopt the strategy of using only 1-hop neighbor information. This simplified input data not only simplifies the model structure but also retains the most direct and relevant local information in the graph. Let $v_i$ denote a node in the graph, and define its 1-hop neighbor within the time interval $[t, t_0]$ as $\mathcal{N}(v_i; t, t_0)$. We introduce an adaptive node feature computation that accounts for the varying neighborhood sizes:

$$s(v_i) = \mathbf{x}(v_i) + \frac{1}{|\mathcal{N}(v_i; t_0 - t, t_0)|} \sum_{v_j \in \mathcal{N}(v_i; t_0 - t, t_0)} \alpha_{ij} \cdot \mathbf{x}(v_j) \tag{11}$$

Here, $\mathbf{x}_i^{\text{node}}$ represents the feature vector of node $v_i$, and $\alpha_{ij}$ is an adaptive weighting factor defined using the standard softmax function:

$$\alpha_{ij} = \frac{\exp(\mathbf{x}(v_{ij}))}{\sum_{v_k \in \mathcal{N}(v_i; t_0 - t, t_0)} \exp(\mathbf{x}(v_{ik}))} \tag{12}$$

After we get temporal link embeddings and node embeddings, we can use them for various graph tasks like link prediction and node classification.

## 4.2 PREDICTION LAYER

For link prediction, we design a link classifier that determines the existence of a link between two nodes at a future time. This classifier utilizes two inputs: (1) $\mathbf{h}(v_i)$, the temporal link embeddings, which capture the temporal link information of node $v_i$, and (2) $s(v_i)$, the node embeddings, which contain the node features and 1-hop neighbor information. We define the representation of node $v_i$, which combines the temporal link embeddings and node embeddings, as the concatenation of these two embeddings:

$$\mathbf{E}(v_i) = [s(v_i) \oplus \mathbf{h}(v_i)] \tag{13}$$

where $\oplus$ denotes vector concatenation. To predict whether an interaction occurs between nodes $v_i$ and $v_j$ at a future time, we employ a two-layer MLP model. This prediction layer takes $\mathbf{E}(v_i)$ and

$\mathbf{E}(v_j)$ as inputs and outputs the probability of a link forming between nodes $v_i$ and $v_j$ at the future time. Formally, we can express this as:

$$P(\text{link}_{ij}) = \text{MLP}(\mathbf{E}(v_i), \mathbf{E}(v_j)) \tag{14}$$

$P(\text{link}_{ij})$ is the likelihood that there exist a link between $v_i$ and $v_j$. Our method considers both node features and much richer dynamic temporal information, enabling a more comprehensive prediction of link formation.

## 5 EXPERIMENTS

### 5.1 DATASET AND BASELINES

Our study includes thirteen diverse datasets, collected by Poursafaei et al. (2022), spanning various domains: Wikipedia, Reddit, MOOC, LastFM, Enron, Social Evolution, UCI, Flights, Canadian Parliament, US Legislature, UN Trade, UN Vote, and Contact. Detailed statistics of these datasets are presented in Table 4. We evaluate our approach against nine state-of-the-art temporal graph learning baselines, representing a broad spectrum of techniques including GNNs, memory networks, random walks, transformers, MLP-mixers, and sequential models: DyGFormer Yu et al. (2023), JODIE Kumar et al. (2019), DyRep Trivedi et al. (2019), TGAT Xu et al. (2020), TGN Rossi et al. (2020), CAWN Wang et al. (2021a), EdgeBank Wang et al. (2021b), TCL Wang et al. (2021b), and GraphMixer Cong et al. (2023).

### 5.2 EVALUATION

We evaluate our model's performance in dynamic link prediction, following established methodologies Yu et al. (2023); Rossi et al. (2020); Wang et al. (2021a). Our task involves predicting the probability of a link forming between two nodes at a specific time, considering both transductive (future links between observed nodes) and inductive (links involving unseen nodes) scenarios. For evaluation, we employ Average Precision (AP) and Area Under the Receiver Operating Characteristic Curve (AUC-ROC) metrics. We adopt random, historical, and inductive negative sampling strategies as described in Poursafaei et al. (2022); Yu et al. (2023). Each dataset is chronologically split into 70% training, 15% validation, and 15% testing sets.

### 5.3 OVERALL PERFORMANCE

We report the performance of different methods on the AP metric for transductive temporal link prediction with three negative sampling strategies (random, historical, and inductive) in Table 1. The **best** results are emphasized by **bold** fonts, and the second-best results are underlined. Please refer to Table 7, Table 8 and Table 9 for the results of AP for inductive dynamic link prediction tasks.

The results demonstrate the consistently high performance of NODE-SAT across various datasets and experimental settings. In the random (rnd) negative sampling strategy setting, NODE-SAT consistently outperformed other methods, achieving perfect 100% accuracy ($\pm 0.00$) on 6 out of 13 datasets, including Wikipedia, Reddit, Social Evolution, Flights, and Can.Parl. Even on challenging datasets like UN Trade and UN Vote, NODE-SAT maintained high accuracy (95.47% $\pm 0.89$ and 83.89% $\pm 3.00$ respectively), significantly surpassing other methods. For the historical (hist) negative sampling strategy setting, NODE-SAT continued its strong performance, achieving 100% accuracy on 5 datasets and over 95% accuracy on 4 others. Notably, it showed remarkable improvement on complex datasets like UN Trade (95.79% $\pm 1.28$) compared to the next best method (EdgeBank at 81.32%). The inductive (ind) negative sampling strategy setting, often considered the most challenging, further highlighted NODE-SAT's capabilities. It maintained 100% accuracy on 3 datasets and achieved over 95% accuracy on 5 others. Across all settings, NODE-SAT consistently outperformed state-of-the-art methods like DyGFormer, GraphMixer, and TGN. Our method's ability to maintain high accuracy with low standard deviations across diverse datasets and settings underscores its reliability and effectiveness in temporal graph learning.

Fig 3, Fig 4, and Fig 5 illustrate the training loss and ROC AUC for the Can.Parl, MOOC, and Wikipedia datasets. NODE-SAT consistently outperforms state-of-the-art methods like TCL,

Table 1: AP for transductive dynamic link prediction with random, historical, and inductive negative sampling strategies. NSS is Negative Sampling Strategies.

| NSS | Datasets | JODIE | DyRep | TGAT | TGN | CAWN | EdgeBank | TCL | GraphMixer | DyGFormer | NODE-SAT |
|---|---|---|---|---|---|---|---|---|---|---|---|
| rnd | Wikipedia | 96.50 ± 0.14 | 94.86 ± 0.06 | 96.94 ± 0.06 | 98.45 ± 0.06 | 98.76 ± 0.03 | 90.37 ± 0.00 | 96.47 ± 0.16 | 97.25 ± 0.03 | 99.03 ± 0.02 | **100 ± 0.00** |
| | Reddit | 98.31 ± 0.14 | 98.22 ± 0.04 | 98.52 ± 0.02 | 98.63 ± 0.06 | 99.11 ± 0.01 | 94.86 ± 0.00 | 97.53 ± 0.02 | 97.31 ± 0.01 | 99.22 ± 0.01 | **100 ± 0.00** |
| | MOOC | 80.23 ± 2.44 | 81.97 ± 0.49 | 85.84 ± 0.15 | 89.15 ± 1.60 | 80.15 ± 0.25 | 57.97 ± 0.00 | 82.38 ± 0.24 | 82.78 ± 0.15 | 87.52 ± 0.49 | **99.50 ± 0.29** |
| | LastFM | 70.85 ± 2.13 | 71.92 ± 2.21 | 73.42 ± 0.21 | 77.07 ± 3.97 | 86.99 ± 0.06 | 79.29 ± 0.00 | 67.27 ± 2.16 | 75.61 ± 0.24 | 93.00 ± 0.12 | **94.92 ± 0.98** |
| | Enron | 84.77 ± 0.30 | 82.38 ± 3.36 | 71.12 ± 0.97 | 86.53 ± 1.11 | 89.56 ± 0.09 | 83.53 ± 0.00 | 79.70 ± 0.71 | 82.25 ± 0.16 | 92.47 ± 0.12 | **98.68 ± 1.58** |
| | Social Evo. | 89.89 ± 0.55 | 88.87 ± 0.30 | 93.16 ± 0.17 | 93.57 ± 0.17 | 84.96 ± 0.09 | 74.95 ± 0.00 | 93.13 ± 0.16 | 93.37 ± 0.07 | 94.73 ± 0.01 | **100 ± 0.00** |
| | UCI | 89.43 ± 1.09 | 65.14 ± 2.30 | 79.63 ± 0.70 | 92.34 ± 1.04 | 95.18 ± 0.06 | 76.20 ± 0.00 | 89.57 ± 1.63 | 93.25 ± 0.57 | 95.79 ± 0.17 | **99.16 ± 0.72** |
| | Flights | 95.60 ± 1.73 | 95.29 ± 0.72 | 94.03 ± 0.18 | 97.95 ± 0.14 | 98.51 ± 0.01 | 89.35 ± 0.00 | 91.23 ± 0.02 | 90.99 ± 0.05 | 98.91 ± 0.01 | **100 ± 0.00** |
| | Can. Parl. | 69.26 ± 0.31 | 66.54 ± 2.76 | 70.73 ± 0.72 | 70.88 ± 2.34 | 69.82 ± 2.34 | 64.55 ± 0.00 | 68.67 ± 2.67 | 77.04 ± 0.46 | 97.36 ± 0.45 | **99.99 ± 0.01** |
| | US Legis. | 75.05 ± 1.52 | 75.34 ± 0.39 | 68.52 ± 3.16 | 75.99 ± 0.58 | 70.58 ± 0.48 | 58.39 ± 0.00 | 69.59 ± 0.48 | 70.74 ± 1.02 | 71.11 ± 0.59 | **86.34 ± 6.52** |
| | UN Trade | 64.94 ± 0.31 | 63.21 ± 0.93 | 61.47 ± 0.18 | 65.03 ± 1.37 | 65.39 ± 0.12 | 60.41 ± 0.00 | 62.21 ± 0.03 | 62.61 ± 0.27 | 66.46 ± 1.29 | **95.47 ± 0.89** |
| | UN Vote | 63.91 ± 0.81 | 62.81 ± 0.80 | 52.21 ± 0.98 | 65.72 ± 2.17 | 52.84 ± 0.10 | 58.49 ± 0.00 | 51.90 ± 0.30 | 52.11 ± 0.16 | 55.55 ± 0.42 | **83.89 ± 3.00** |
| | Contact | 95.31 ± 1.33 | 95.98 ± 0.15 | 96.28 ± 0.09 | 96.89 ± 0.56 | 90.26 ± 0.28 | 92.58 ± 0.00 | 92.44 ± 0.12 | 91.92 ± 0.03 | 98.29 ± 0.01 | **100 ± 0.00** |
| hist | Wikipedia | 97.37 ± 0.07 | 97.13 ± 0.07 | 97.73 ± 0.03 | 98.67 ± 0.04 | 98.89 ± 0.02 | 98.71 ± 0.00 | 97.39 ± 0.11 | 97.99 ± 0.02 | 99.14 ± 0.01 | **100 ± 0.00** |
| | Reddit | 98.70 ± 0.09 | 98.77 ± 0.02 | 98.91 ± 0.01 | 99.01 ± 0.03 | 99.29 ± 0.01 | 99.52 ± 0.00 | 98.35 ± 0.02 | 98.13 ± 0.01 | 99.38 ± 0.01 | **100 ± 0.00** |
| | MOOC | 84.51 ± 1.26 | 86.41 ± 0.30 | 89.29 ± 0.15 | 91.88 ± 0.97 | 84.21 ± 0.24 | 84.66 ± 0.00 | 86.95 ± 0.20 | 87.01 ± 0.13 | 90.68 ± 0.37 | **99.83 ± 0.17** |
| | LastFM | 88.68 ± 1.01 | 88.56 ± 1.23 | 90.06 ± 0.13 | 92.42 ± 1.99 | 94.38 ± 0.04 | 97.52 ± 0.00 | 87.56 ± 1.16 | 91.69 ± 0.14 | 97.16 ± 0.07 | **98.04 ± 1.03** |
| | Enron | 89.77 ± 0.18 | 89.19 ± 1.87 | 81.32 ± 0.62 | 91.63 ± 0.65 | 93.16 ± 0.06 | 95.58 ± 0.00 | 87.32 ± 0.44 | 89.31 ± 0.11 | 95.27 ± 0.08 | **99.01 ± 1.32** |
| | Social Evo. | 91.59 ± 0.36 | 91.48 ± 0.20 | 94.55 ± 0.11 | 94.89 ± 0.11 | 88.47 ± 0.07 | 92.02 ± 0.00 | 94.74 ± 0.11 | 94.87 ± 0.05 | 95.87 ± 0.01 | **100 ± 0.00** |
| | UCI | 93.76 ± 0.60 | 79.05 ± 1.47 | 89.08 ± 0.44 | 95.70 ± 0.60 | 97.13 ± 0.04 | 94.13 ± 0.00 | 94.23 ± 0.92 | 96.39 ± 0.33 | 97.55 ± 0.10 | **99.14 ± 0.23** |
| | Flights | 96.95 ± 0.96 | 96.98 ± 0.40 | 96.30 ± 0.10 | 98.66 ± 0.08 | 98.94 ± 0.01 | 98.07 ± 0.00 | 94.54 ± 0.01 | 94.36 ± 0.03 | 99.24 ± 0.01 | **100 ± 0.00** |
| | Can. Parl. | 78.80 ± 0.20 | 77.52 ± 1.72 | 80.35 ± 0.45 | 80.46 ± 1.45 | 79.68 ± 1.45 | 84.91 ± 0.00 | 79.17 ± 1.66 | 84.73 ± 0.29 | 98.55 ± 0.27 | **100 ± 0.00** |
| | US Legis. | 83.69 ± 0.92 | 84.40 ± 0.24 | 79.62 ± 1.93 | 84.92 ± 0.35 | 81.19 ± 0.29 | 81.03 ± 0.00 | 80.72 ± 0.29 | 81.57 ± 0.62 | 81.83 ± 0.36 | **92.15 ± 6.35** |
| | UN Trade | 71.49 ± 0.19 | 70.39 ± 0.57 | 69.22 ± 0.11 | 71.55 ± 0.84 | 71.78 ± 0.07 | 73.62 ± 0.00 | 69.78 ± 0.02 | 70.07 ± 0.17 | 72.95 ± 0.79 | **97.76 ± 0.46** |
| | UN Vote | 73.02 ± 0.51 | 72.39 ± 0.50 | 64.47 ± 0.61 | 74.25 ± 1.35 | 65.02 ± 0.06 | 76.05 ± 0.00 | 64.07 ± 0.19 | 64.22 ± 0.10 | 67.11 ± 0.26 | **89.64 ± 2.51** |
| | Contact | 96.62 ± 0.75 | 97.07 ± 0.08 | 97.26 ± 0.05 | 97.67 ± 0.31 | 93.08 ± 0.16 | 98.21 ± 0.00 | 94.74 ± 0.07 | 94.40 ± 0.02 | 98.89 ± 0.01 | **100 ± 0.00** |
| ind | Wikipedia | 96.34 ± 0.15 | 93.66 ± 0.06 | 96.44 ± 0.06 | 98.11 ± 0.07 | 98.47 ± 0.03 | 86.49 ± 0.00 | 95.81 ± 0.18 | 96.70 ± 0.03 | 98.77 ± 0.02 | **100 ± 0.00** |
| | Reddit | 98.03 ± 0.16 | 97.86 ± 0.05 | 98.21 ± 0.02 | 98.34 ± 0.07 | 98.92 ± 0.01 | 92.51 ± 0.00 | 97.01 ± 0.02 | 96.77 ± 0.01 | 99.05 ± 0.01 | **100 ± 0.00** |
| | MOOC | 77.62 ± 2.70 | 79.10 ± 0.55 | 83.69 ± 0.17 | 87.32 ± 1.77 | 77.59 ± 0.27 | 49.96 ± 0.00 | 79.55 ± 0.27 | 80.08 ± 0.17 | 85.46 ± 0.54 | **99.33 ± 0.38** |
| | LastFM | 63.97 ± 2.37 | 64.89 ± 2.45 | 67.04 ± 0.23 | 71.15 ± 4.39 | 83.67 ± 0.07 | 70.93 ± 0.00 | 59.39 ± 2.39 | 68.70 ± 0.27 | 91.04 ± 0.13 | **91.89 ± 2.13** |
| | Enron | 81.98 ± 0.33 | 78.63 ± 3.72 | 65.15 ± 1.07 | 83.51 ± 1.23 | 87.61 ± 0.10 | 76.91 ± 0.00 | 75.30 ± 0.79 | 78.15 ± 0.18 | 90.90 ± 0.13 | **97.51 ± 3.29** |
| | Social Evo. | 88.89 ± 0.62 | 87.57 ± 0.34 | 92.37 ± 0.19 | 92.81 ± 0.19 | 82.93 ± 0.10 | 66.98 ± 0.00 | 92.17 ± 0.18 | 92.46 ± 0.08 | 94.04 ± 0.01 | **100 ± 0.00** |
| | UCI | 86.96 ± 1.21 | 58.30 ± 2.55 | 74.63 ± 0.78 | 90.39 ± 1.15 | 93.99 ± 0.07 | 67.48 ± 0.00 | 86.89 ± 1.80 | 91.37 ± 0.63 | 94.71 ± 0.19 | **97.14 ± 0.58** |
| | Flights | 94.77 ± 1.92 | 94.38 ± 0.80 | 92.84 ± 0.20 | 97.51 ± 0.15 | 98.24 ± 0.01 | 84.84 ± 0.00 | 89.35 ± 0.02 | 89.07 ± 0.06 | 98.71 ± 0.01 | **100 ± 0.00** |
| | Can. Parl. | 64.13 ± 0.34 | 60.95 ± 3.06 | 65.55 ± 0.80 | 65.72 ± 2.59 | 64.57 ± 2.59 | 55.10 ± 0.00 | 63.07 ± 2.96 | 72.83 ± 0.51 | 96.66 ± 0.50 | **99.99 ± 0.01** |
| | US Legis. | 70.38 ± 1.68 | 70.27 ± 0.43 | 62.41 ± 3.50 | 71.08 ± 0.64 | 64.87 ± 0.53 | 48.11 ± 0.00 | 63.50 ± 0.53 | 64.83 ± 1.13 | 65.24 ± 0.65 | **87.28 ± 6.50** |
| | UN Trade | 61.41 ± 0.34 | 59.45 ± 1.03 | 57.42 ± 0.20 | 61.51 ± 1.52 | 61.91 ± 0.13 | 54.16 ± 0.00 | 58.26 ± 0.03 | 58.71 ± 0.30 | 63.10 ± 1.43 | **94.32 ± 1.11** |
| | UN Vote | 59.15 ± 0.90 | 57.93 ± 0.89 | 46.21 ± 1.09 | 61.18 ± 2.41 | 46.91 ± 0.11 | 50.36 ± 0.00 | 45.87 ± 0.33 | 46.10 ± 0.18 | 49.93 ± 0.47 | **76.77 ± 5.14** |
| | Contact | 94.51 ± 1.47 | 95.30 ± 0.17 | 95.65 ± 0.10 | 96.36 ± 0.62 | 88.48 ± 0.31 | 89.31 ± 0.00 | 91.05 ± 0.13 | 90.47 ± 0.03 | 97.89 ± 0.01 | **100 ± 0.00** |

JODIE, TGAT, TGN, and DyGformer across multiple metrics. It achieves the lowest training loss and highest ROC AUC scores, particularly on the MOOC and Can.Parl datasets, with ROC AUC approaching 1.0 on Can.Parl. NODE-SAT demonstrates accelerated convergence, reaching stable performance in fewer epochs than its counterparts, while maintaining exceptional stability with minimal fluctuations. Its consistent high performance across datasets of varying sizes and complexities underscores its scalability and robustness in temporal graph learning.

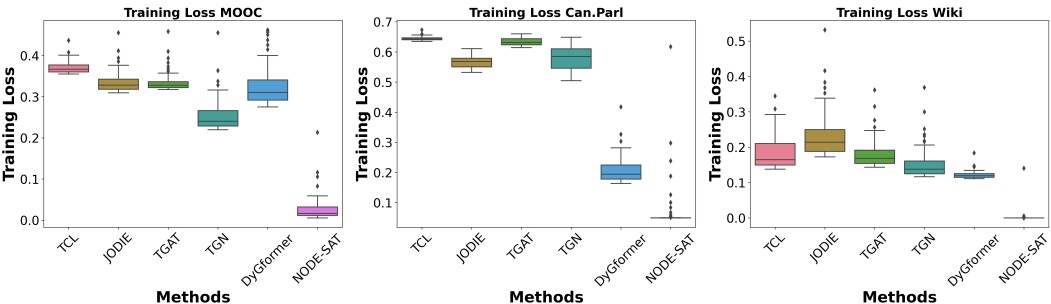

Figure 3: Training Loss Comparison

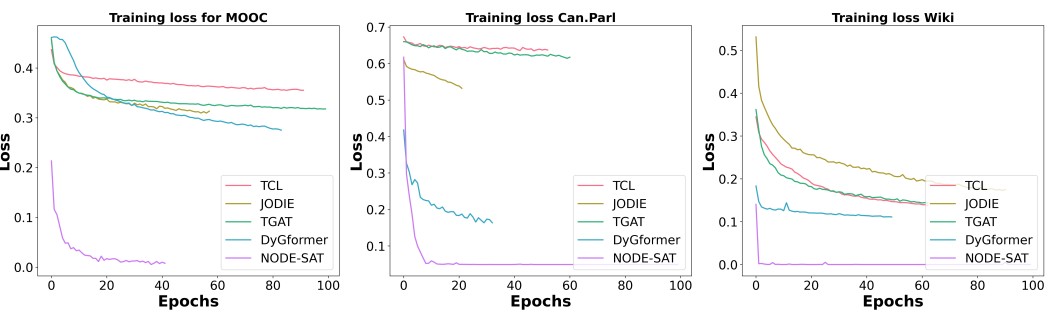

Figure 4: Training Loss Comparison

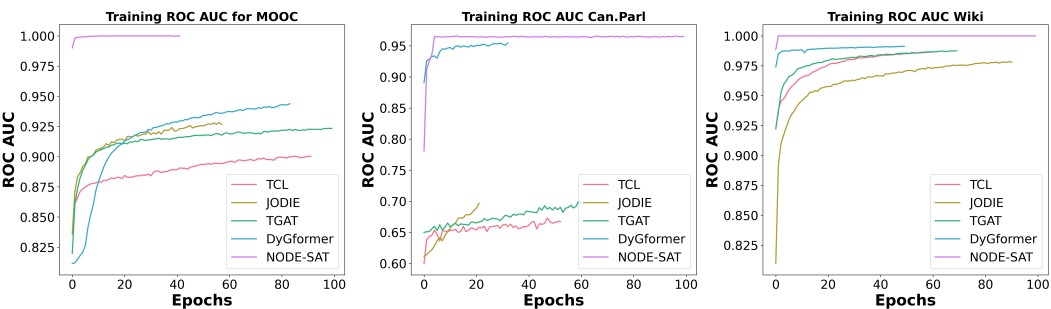

Figure 5: Training ROC AUC Comparison

## 6 ABLATION STUDY

### 6.1 EMPIRICAL EVALUATION OF NODE INTEGRATION IN NODE-SAT

We are interested in investigating the impact of incorporating NODE on our model's performance across various temporal graph datasets. Our study aims to assess how NODE influences the effectiveness of NODE-SAT. To quantify this effect, we conducted a comparative analysis between two versions of our model: NODE-SAT with NODE integration and NODE-SAT without NODE. Table 2 presents the results of this performance comparison. The performance comparison between

Table 2: Performance Comparison: NODE-SAT with and without NODE

| Datasets | Without NODE | With NODE |
|---|---|---|
| Wikipedia | $99.99 \pm 0.00$ | $\mathbf{100 \pm 0.00}$ |
| Reddit | $100 \pm 0.00$ | $100 \pm 0.00$ |
| MOOC | $99.51 \pm 0.11$ | $\mathbf{99.61 \pm 0.20}$ |
| LastFM | $89.65 \pm 2.41$ | $\mathbf{94.92 \pm 0.98}$ |
| Enron | $73.61 \pm 6.48$ | $\mathbf{98.68 \pm 1.58}$ |
| Soc. Evol. | $100 \pm 0.00$ | $100 \pm 0.00$ |
| UCI | $97.48 \pm 0.79$ | $\mathbf{99.16 \pm 0.72}$ |
| Flights | $100 \pm 0.00$ | $100 \pm 0.00$ |
| Can. Parl. | $98.29 \pm 3.12$ | $\mathbf{99.99 \pm 0.01}$ |
| US Leg. | $77.89 \pm 14.11$ | $\mathbf{86.34 \pm 6.52}$ |
| UN Trade | $63.05 \pm 5.13$ | $\mathbf{95.47 \pm 0.89}$ |
| UN Vote | $54.60 \pm 0.57$ | $\mathbf{83.89 \pm 3.00}$ |
| Contact | $100 \pm 0.00$ | $100 \pm 0.00$ |

NODE-SAT with and without NODE reveals significant benefits of incorporating NODE into our model. Across the 13 datasets tested, NODE integration either maintains or improves performance in all cases. The most dramatic improvements are observed in the UN Trade and UN Vote datasets, with increases of 32.42 and 29.29 percentage points respectively. Substantial enhancements are also seen in the Enron dataset (25.07 points) and LastFM (5.27 points). NODE integration often leads to increased stability, as evidenced by reduced standard deviations in datasets like Enron (from 6.48 to 1.58) and UN Trade (from 5.13 to 0.89). While some datasets (Wikipedia, Reddit, Social Evolution, Flights, and Contact) already achieve optimal or near-optimal performance without NODE, its integration either maintains this high performance or slightly improves it, as in the case of Wikipedia. Notably, in datasets with lower initial performance, such as US Legis. and UN Vote, NODE-SAT demonstrates substantial improvements while also reducing variability. This consistent enhancement across diverse dataset types strongly supports the integration of NODE in temporal graph models, demonstrating its effectiveness in capturing complex temporal dynamics.

## 6.2 TIME PARAMETER $\tau$ INFLUENCE IN NODE-SAT

We investigate the impact of the time parameter $\tau$ on NODE-SAT's performance across various datasets (Table 3). The results demonstrate NODE-SAT's stability across various $\tau$ values (1.0, 1.3,

Table 3: NODE-SAT Performance Across Different $\tau$ Values

| Datasets | $\tau = 1$ | $\tau = 1.3$ | $\tau = 1.5$ | $\tau = 1.7$ | $\tau = 2.0$ |
|---|---|---|---|---|---|
| Wikipedia | $\mathbf{100 \pm 0.00}$ | $100 \pm 0.00$ | $100 \pm 0.00$ | $100 \pm 0.00$ | $100 \pm 0.00$ |
| Reddit | $\mathbf{100 \pm 0.00}$ | $100 \pm 0.00$ | $100 \pm 0.00$ | $100 \pm 0.00$ | $100 \pm 0.00$ |
| MOOC | $99.50 \pm 0.29$ | $99.55 \pm 0.24$ | $\mathbf{99.61 \pm 0.20}$ | $99.51 \pm 0.20$ | $99.52 \pm 0.14$ |
| LastFM | $93.89 \pm 1.92$ | $\mathbf{94.92 \pm 0.98}$ | $90.64 \pm 2.56$ | $93.52 \pm 3.53$ | $93.43 \pm 2.00$ |
| Enron | $98.01 \pm 2.63$ | $98.68 \pm 1.58$ | $\mathbf{98.92 \pm 1.20}$ | $98.28 \pm 1.49$ | $98.87 \pm 0.68$ |
| Social Evo. | $\mathbf{100 \pm 0.00}$ | $100 \pm 0.00$ | $100 \pm 0.00$ | $100 \pm 0.00$ | $100 \pm 0.00$ |
| UCI | $97.86 \pm 0.46$ | $99.16 \pm 0.72$ | $99.35 \pm 0.36$ | $99.21 \pm 0.44$ | $\mathbf{99.51 \pm 0.29}$ |
| Flights | $\mathbf{100 \pm 0.00}$ | $100 \pm 0.00$ | $100 \pm 0.00$ | $100 \pm 0.00$ | $100 \pm 0.00$ |
| Can. Parl. | $99.99 \pm 0.01$ | $\mathbf{100 \pm 0.01}$ | $100 \pm 0.01$ | $99.99 \pm 0.02$ | $99.99 \pm 0.03$ |
| US Legis. | $\mathbf{86.34 \pm 6.52}$ | $60.68 \pm 21.98$ | $60.77 \pm 24.04$ | $61.39 \pm 30.20$ | $69.80 \pm 26.05$ |
| UN Trade | $\mathbf{95.47 \pm 0.89}$ | $92.62 \pm 2.28$ | $93.11 \pm 3.13$ | $91.94 \pm 0.61$ | $92.44 \pm 3.42$ |
| UN Vote | $81.06 \pm 4.63$ | $83.89 \pm 3.00$ | $83.27 \pm 3.78$ | $84.06 \pm 4.67$ | $\mathbf{84.64 \pm 3.29}$ |
| Contact | $\mathbf{100 \pm 0.00}$ | $100 \pm 0.00$ | $100 \pm 0.00$ | $100 \pm 0.00$ | $100 \pm 0.00$ |

1.5, 1.7, and 2.0). Higher $\tau$ values allow the temporal link embeddings to evolve over a longer time span, capturing more extended temporal dynamics, while lower $\tau$ values focus on shorter-term changes. Five out of 13 datasets (Wikipedia, Reddit, Social Evolution, Flights, and Contact) achieve perfect 100% accuracy ($\pm 0.0000$) for all $\tau$ values, while several others (MOOC, Enron, UCI, and Canadian Parliament) consistently perform above 97%. Some datasets exhibit sensitivity to $\tau$: LastFM's accuracy ranges from 90.64% to 94.92% (best at $\tau$=1.3), US Legis. shows high variability (60.68% to 86.34%, best at $\tau$=1), UN Trade peaks at $\tau$=1 (95.47%), and UN Vote improves slightly with increasing $\tau$ (best at $\tau$=2.0). While $\tau$=1 often yields optimal or near-optimal results, the best $\tau$ value appears dataset-dependent. NODE-SAT's ability to maintain high accuracy across various $\tau$ values for most datasets underscores its effectiveness in capturing temporal dynamics in diverse graph datasets, though careful tuning may be beneficial for more complex cases. The optimal choice of this parameter of NODE-SAT can be influenced by the specific temporal graph dataset. Future work could explore the relationship between dataset properties and optimal $\tau$ values to develop guidelines for parameter selection in different domains.

## 7 CONCLUSION

We introduce NODE-SAT, a novel temporal graph learning model that integrates Neural Ordinary Differential Equations (NODEs) with self-attention mechanisms. Through extensive experiments on thirteen diverse datasets, NODE-SAT consistently demonstrates outstanding performance in temporal link prediction tasks, exhibiting exceptional accuracy, rapid convergence, robustness across varying dataset complexities, and strong generalization capabilities in both transductive and inductive temporal link prediction settings. The model's innovative combination of NODEs and self-attention provides a simple yet powerful framework for temporal graph modeling, allowing for nuanced representation of graph evolution and enhanced capture of complex between-node relationships. By leveraging a continuous-time perspective, NODE-SAT effectively models node correlations and temporal link dynamics, potentially discerning subtle patterns that discrete-time models might overlook. These results not only validate NODE-SAT's efficacy in temporal graph learning tasks but also open up new research directions in continuous-time graph modeling.

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

# A  APPENDIX

## A.1  DATASET DETAIL

The thirteen datasets used in our experiments exhibit diverse characteristics, providing a comprehensive testbed for our model. They range in size from 74 nodes (Social Evolution) to 10,984 nodes (Reddit), with edge counts varying from 51,146 (Contact) to 1,873,731 (Social Evolution). Time spans covered by these datasets are equally varied, spanning from 1 month to 230 years, allowing for evaluation of both short-term and long-term temporal dynamics. Edge feature counts range from 0 to 172, with Reddit and Wikipedia offering the richest feature sets. Network densities show significant variation, from very sparse (Reddit and Wikipedia with densities near 0) to extremely dense (Social Evolution with a density of 346.86). Average node degrees also vary widely, from 34.13 (Wikipedia) to 50,641.38 (Social Evolution), indicating diverse connectivity patterns.

## A.2  IMPLEMENTATION DETAILS

We optimize all models using binary cross-entropy loss as the objective function. We train the models for 100 epochs and apply an early stopping strategy with a patience of 20. We select the model that achieves the best performance on the validation set for testing. The learning rate and batch size are set to 0.0001 and 200, respectively, for all methods across all datasets. We run the methods

Table 4: Statistics of the datasets.

| Datasets | Domains | #Nodes | #Links | Bipartite | Duration | Unique Steps |
|----------|---------|--------|--------|-----------|----------|--------------|
| Wikipedia | Social | 9,227 | 157,474 | True | 1 month | 152,757 |
| Reddit | Social | 10,984 | 672,447 | True | 1 month | 669,065 |
| MOOC | Interaction | 7,144 | 411,749 | True | 17 months | 345,600 |
| LastFM | Interaction | 1,980 | 1,293,103 | True | 1 month | 1,283,614 |
| Enron | Social | 184 | 125,235 | False | 3 years | 22,632 |
| Social Evo. | Proximity | 74 | 2,099,519 | False | 8 months | 565,932 |
| UCI | Social | 1,899 | 59,835 | False | 196 days | 58,911 |
| Flights | Transport | 13,169 | 1,927,145 | False | 4 months | 122 |
| Can. Parl. | Politics | 734 | 74,478 | False | 14 years | 14 |
| US Legis. | Politics | 225 | 60,396 | False | 12 congresses | 12 |
| UN Trade | Economics | 255 | 507,497 | False | 32 years | 32 |
| UN Vote | Politics | 201 | 1,035,742 | False | 72 years | 72 |
| Contact | Proximity | 692 | 2,426,279 | False | 1 month | 8,064 |

five times with seeds from 0 to 4 and report the average performance to minimize deviations. Experiments are conducted on an Ubuntu machine equipped with one AMD Ryzen 9 7950X 16-Core Processor. The GPU device is an NVIDIA RTX 4090.

## A.3 BASELINES

## A.4 PERFORMANCE HEATMAP FOR TRANSDUCTIVE TEMPORAL LINK PREDICTION

We provide performance heatmaps for transductive temporal link prediction in Fig. 6, Fig. 7, and Fig. 8.

## A.5 AP FOR INDUCTIVE DYNAMIC LINK PREDICTION

Tables 7, 8, and 9 present the Average Precision (AP) results for Inductive Dynamic Link Prediction using Random, Historical, and Inductive Negative Sampling, respectively. The tables compare nine models (JODIE, DyRep, TGAT, TGN, CAWN, TCL, GraphMixer, DyGFormer, and NODE-SAT) across various datasets. NODE-SAT consistently outperforms other models in most scenarios, achieving perfect 100% AP on several datasets, particularly in the Random Negative Sampling setting. It maintains strong performance in Historical and Inductive settings, though with slightly lower scores on some datasets. Notably, NODE-SAT shows remarkable improvement on challenging datasets like UN Trade and UN Vote. However, it underperforms on the US Legislature dataset across all settings. Other models, particularly DyGFormer and GraphMixer, often emerge as strong contenders, frequently achieving the second-best scores. The results demonstrate NODE-SAT's overall superiority in Inductive Dynamic Link Prediction tasks, with some specific exceptions, highlighting its effectiveness across different negative sampling strategies and diverse datasets.

Table 5: Dataset Details

| Dataset | Description |
|---|---|
| Wikipedia | A bipartite interaction graph of edits on Wikipedia pages over one month. Nodes represent users and pages, links denote editing behaviors with timestamps. Each link has a 172-dimensional LIWC feature. Includes dynamic labels indicating temporary user bans. |
| Reddit | A bipartite graph recording user posts under subreddits during one month. Users and subreddits are nodes, links are timestamped posting requests. Each link has a 172-dimensional LIWC feature. Includes dynamic labels for user bans. |
| MOOC | A bipartite interaction network of online sources. Nodes are students and course content units. Links denote student access to content, with 4-dimensional features. |
| LastFM | A bipartite graph of user listening behaviors over one month. Users and songs are nodes, links represent listening events. |
| Enron | Records email communications between ENRON energy corporation employees over three years. |
| Social Evo. | A mobile phone proximity network monitoring undergraduate dormitory activities for eight months. Links have 2-dimensional features. |
| UCI | An online communication network where nodes are university students and links are messages posted by students. |
| Flights | A dynamic flight network showing air traffic development during the COVID-19 pandemic. Nodes are airports, links are tracked flights with weights indicating daily flight numbers. |
| Can. Parl. | A dynamic political network recording interactions between Canadian MPs from 2006 to 2019. Nodes are MPs, links created when two MPs vote "yes" on a bill. Link weights show yearly co-voting counts. |
| US Legis. | A senate co-sponsorship network tracking social interactions between US legislators. Link weights indicate bill co-sponsorship counts per congress. |
| UN Trade | Contains food and agriculture trade between 181 nations for over 30 years. Link weights show normalized agriculture import/export values between countries. |
| UN Vote | Records roll-call votes in the UN General Assembly. Link weights increase when two nations both vote "yes" on an item. |
| Contact | Describes physical proximity evolution among about 700 university students over a month. Links denote close proximity, with weights indicating proximity levels. |

Table 6: Descriptions of Baselines

| Baseline | Description |
|---|---|
| CAWN | A continuous-time model that utilizes a novel time encoding method and MLP-based feature processing. It implements an attention mechanism across multiple time windows to effectively capture temporal patterns in dynamic graphs. |
| TGN | A dynamic graph learning framework featuring a memory module for long-term dependency capture. TGN generates temporal node embeddings through a combination of message passing and memory update mechanisms. |
| JODIE | An approach using coupled recurrent neural networks to learn dynamic node embeddings. JODIE is designed to predict future interactions and node trajectories in dynamic graphs, employing separate RNNs for updating user and item embeddings. |
| DyRep | A deep recurrent model designed to capture both topological and temporal dependencies in dynamic graphs. It employs a two-time-scale framework to simultaneously model structural evolution and node dynamics. |
| TGAT | An extension of graph attention mechanisms to temporal settings. TGAT incorporates temporal information into node embeddings using advanced time-encoding techniques, enhancing the model's ability to handle time-varying graph data. |

Table 7: AP for Inductive Dynamic Link Prediction with Random Negative Sampling (Best Scores in Bold)

| Datasets | JODIE | DyRep | TGAT | TGN | CAWN | TCL | GraphMixer | DyGFormer | NODE-SAT |
|---|---|---|---|---|---|---|---|---|---|
| Wikipedia | 94.82 ± 0.20 | 92.43 ± 0.37 | 96.22 ± 0.07 | 97.83 ± 0.04 | 98.24 ± 0.03 | 96.22 ± 0.17 | 96.65 ± 0.02 | 98.59 ± 0.03 | **100 ± 0.00** |
| Reddit | 96.50 ± 0.13 | 96.09 ± 0.11 | 97.09 ± 0.04 | 97.50 ± 0.07 | 98.62 ± 0.01 | 94.09 ± 0.07 | 95.26 ± 0.02 | 98.84 ± 0.02 | **100 ± 0.00** |
| MOOC | 79.63 ± 1.92 | 81.07 ± 0.44 | 85.50 ± 0.19 | 89.04 ± 1.17 | 81.42 ± 0.24 | 80.60 ± 0.22 | 81.41 ± 0.21 | 86.96 ± 0.43 | **99.29 ± 0.16** |
| LastFM | 81.61 ± 3.82 | 83.02 ± 1.48 | 78.63 ± 0.31 | 81.45 ± 4.29 | 89.42 ± 0.07 | 73.53 ± 1.66 | 82.11 ± 0.42 | 94.23 ± 0.09 | **94.88 ± 1.71** |
| Enron | 80.72 ± 1.39 | 74.55 ± 3.95 | 67.05 ± 1.51 | 77.94 ± 1.02 | 86.35 ± 0.51 | 76.14 ± 0.79 | 75.88 ± 0.48 | 89.76 ± 0.34 | **97.07 ± 2.06** |
| Social Evo. | 91.96 ± 0.48 | 90.04 ± 0.47 | 91.41 ± 0.16 | 90.77 ± 0.86 | 79.94 ± 0.18 | 91.55 ± 0.09 | 91.86 ± 0.06 | 93.14 ± 0.04 | **100 ± 0.00** |
| UCI | 79.86 ± 1.48 | 57.48 ± 1.87 | 79.54 ± 0.48 | 88.12 ± 2.05 | 92.73 ± 0.06 | 87.36 ± 2.03 | 91.19 ± 0.42 | 94.54 ± 0.12 | **99.21 ± 0.34** |
| Flights | 94.74 ± 0.37 | 92.88 ± 0.73 | 88.73 ± 0.33 | 95.03 ± 0.60 | 97.06 ± 0.02 | 83.41 ± 0.07 | 83.03 ± 0.05 | 97.79 ± 0.02 | **99.79 ± 0.11** |
| Can. Parl. | 53.92 ± 0.94 | 54.02 ± 0.76 | 55.18 ± 0.79 | 54.10 ± 0.93 | 55.80 ± 0.69 | 54.30 ± 0.66 | 55.91 ± 0.82 | 87.74 ± 0.71 | **97.34 ± 1.49** |
| US Legis. | 54.93 ± 2.29 | 57.28 ± 0.71 | 51.00 ± 3.11 | **58.63 ± 0.37** | 53.17 ± 1.20 | 52.59 ± 0.97 | 50.71 ± 0.76 | 54.28 ± 2.87 | 54.58 ± 1.78 |
| UN Trade | 59.65 ± 0.77 | 57.02 ± 0.69 | 61.03 ± 0.18 | 58.31 ± 3.15 | 65.24 ± 0.21 | 62.21 ± 0.12 | 62.17 ± 0.31 | 64.55 ± 0.62 | **78.52 ± 5.71** |
| UN Vote | 56.64 ± 0.96 | 54.62 ± 2.22 | 52.24 ± 1.46 | 58.85 ± 2.51 | 49.94 ± 0.45 | 51.60 ± 0.97 | 50.68 ± 0.44 | 55.93 ± 0.39 | **78.12 ± 3.53** |
| Contact | 94.34 ± 1.45 | 92.18 ± 0.41 | 95.87 ± 0.11 | 93.82 ± 0.99 | 89.55 ± 0.30 | 91.11 ± 0.12 | 90.59 ± 0.05 | 98.03 ± 0.02 | **100.0 ± 0.00** |

Table 8: AP for Inductive Dynamic Link Prediction with Historical Negative Sampling (Best Scores in Bold)

| Datasets | JODIE | DyRep | TGAT | TGN | CAWN | TCL | GraphMixer | DyGFormer | NODE-SAT |
|---|---|---|---|---|---|---|---|---|---|
| Wikipedia | 68.69 ± 0.39 | 62.18 ± 1.27 | 84.17 ± 0.22 | 81.76 ± 0.32 | 67.27 ± 1.63 | 82.20 ± 2.18 | 87.60 ± 0.30 | 71.42 ± 4.43 | **100 ± 0.00** |
| Reddit | 62.34 ± 0.54 | 61.60 ± 0.72 | 63.47 ± 0.36 | 64.85 ± 0.85 | 63.67 ± 0.41 | 60.83 ± 0.25 | 64.50 ± 0.26 | 65.37 ± 0.60 | **100 ± 0.00** |
| MOOC | 63.22 ± 1.55 | 62.93 ± 1.24 | 76.73 ± 0.29 | 77.07 ± 3.41 | 74.68 ± 0.68 | 74.27 ± 0.53 | 74.00 ± 0.97 | 80.82 ± 0.30 | **99.24 ± 0.43** |
| LastFM | 70.39 ± 4.31 | 71.45 ± 1.76 | 76.27 ± 0.25 | 66.65 ± 6.11 | 71.33 ± 0.47 | 65.78 ± 0.65 | 76.42 ± 0.22 | 76.35 ± 0.52 | **97.34 ± 0.97** |
| Enron | 65.86 ± 3.71 | 62.08 ± 2.27 | 61.40 ± 1.31 | 62.91 ± 1.16 | 60.70 ± 0.36 | 67.11 ± 0.62 | 72.37 ± 1.37 | 67.07 ± 0.62 | **97.09 ± 0.41** |
| Social Evo. | 88.51 ± 0.87 | 88.72 ± 1.10 | 93.97 ± 0.54 | 90.66 ± 1.62 | 79.83 ± 0.38 | 94.10 ± 0.31 | 94.01 ± 0.47 | 96.82 ± 0.16 | **100 ± 0.00** |
| UCI | 63.11 ± 2.27 | 52.47 ± 2.06 | 70.52 ± 0.93 | 70.78 ± 0.78 | 64.54 ± 0.47 | 76.71 ± 1.00 | 81.66 ± 0.49 | 72.13 ± 1.87 | **99.51 ± 0.36** |
| Flights | 61.01 ± 1.65 | 62.83 ± 1.31 | 64.72 ± 0.36 | 59.31 ± 1.43 | 56.82 ± 0.57 | 64.50 ± 0.25 | 65.28 ± 0.24 | 57.11 ± 0.21 | **99.90 ± 0.08** |
| Can. Parl. | 52.60 ± 0.88 | 52.28 ± 0.31 | 56.72 ± 0.47 | 54.42 ± 0.77 | 57.14 ± 0.07 | 55.71 ± 0.74 | 55.84 ± 0.73 | 87.40 ± 0.85 | **93.62 ± 8.44** |
| US Legis. | 52.94 ± 2.11 | **62.10 ± 1.41** | 51.83 ± 3.95 | 61.18 ± 1.10 | 55.56 ± 1.71 | 53.87 ± 1.41 | 52.03 ± 1.02 | 56.31 ± 3.46 | 38.22 ± 3.84 |
| UN Trade | 55.46 ± 1.19 | 55.49 ± 0.84 | 55.28 ± 0.71 | 52.80 ± 3.19 | 55.00 ± 0.38 | 55.76 ± 1.03 | 54.94 ± 0.97 | 53.20 ± 1.07 | **84.56 ± 1.94** |

Table 9: AP for Inductive Dynamic Link Prediction with Inductive Negative Sampling (Best Scores in Bold)

| Datasets | JODIE | DyRep | TGAT | TGN | CAWN | TCL | GraphMixer | DyGFormer | NODE-SAT |
|---|---|---|---|---|---|---|---|---|---|
| Wikipedia | 68.70 ± 0.39 | 62.19 ± 1.28 | 84.17 ± 0.22 | 81.77 ± 0.32 | 67.24 ± 1.63 | 82.20 ± 2.18 | 87.60 ± 0.29 | 71.42 ± 4.43 | **100 ± 0.00** |
| Reddit | 62.32 ± 0.54 | 61.58 ± 0.72 | 63.40 ± 0.36 | 64.84 ± 0.84 | 63.65 ± 0.41 | 60.81 ± 0.26 | 64.49 ± 0.25 | 65.35 ± 0.60 | **100 ± 0.00** |
| MOOC | 63.22 ± 1.55 | 62.92 ± 1.24 | 76.72 ± 0.30 | 77.07 ± 3.40 | 74.69 ± 0.68 | 74.28 ± 0.53 | 73.99 ± 0.97 | 80.82 ± 0.30 | **99.24 ± 0.43** |
| LastFM | 70.39 ± 4.31 | 71.45 ± 1.75 | 76.28 ± 0.25 | 69.46 ± 4.65 | 71.33 ± 0.47 | 65.78 ± 0.65 | 76.42 ± 0.22 | 76.35 ± 0.52 | **97.34 ± 0.97** |
| Enron | 65.86 ± 3.71 | 62.08 ± 2.27 | 61.40 ± 1.30 | 62.90 ± 1.16 | 60.72 ± 0.36 | 67.11 ± 0.62 | 72.37 ± 1.38 | 67.07 ± 0.62 | **97.09 ± 0.41** |
| Social Evo. | 88.51 ± 0.87 | 88.72 ± 1.10 | 93.97 ± 0.54 | 90.65 ± 1.62 | 79.83 ± 0.38 | 94.10 ± 0.31 | 94.01 ± 0.47 | 96.82 ± 0.16 | **100 ± 0.00** |
| UCI | 63.16 ± 2.27 | 52.47 ± 2.06 | 70.49 ± 0.93 | 70.73 ± 0.78 | 64.54 ± 0.47 | 76.65 ± 1.00 | 81.64 ± 0.49 | 72.13 ± 1.87 | **99.51 ± 0.36** |
| Flights | 61.01 ± 1.65 | 62.83 ± 1.31 | 64.72 ± 0.36 | 59.32 ± 1.43 | 56.82 ± 0.57 | 64.50 ± 0.25 | 65.29 ± 0.24 | 57.11 ± 0.21 | **99.90 ± 0.08** |
| Can. Parl. | 52.58 ± 0.88 | 52.24 ± 0.31 | 56.46 ± 0.47 | 54.18 ± 0.77 | 57.06 ± 0.07 | 55.46 ± 0.74 | 55.76 ± 0.73 | 87.22 ± 0.85 | **93.62 ± 8.44** |
| US Legis. | 52.94 ± 2.11 | **62.10 ± 1.41** | 51.83 ± 3.95 | 61.18 ± 1.10 | 55.56 ± 1.71 | 53.87 ± 1.41 | 52.03 ± 1.02 | 56.31 ± 3.46 | 43.91 ± 7.12 |
| UN Trade | 55.43 ± 1.19 | 55.42 ± 0.84 | 55.58 ± 0.71 | 52.80 ± 3.19 | 54.97 ± 0.38 | 55.66 ± 1.03 | 54.88 ± 0.97 | 52.56 ± 1.07 | **84.56 ± 1.94** |
| UN Vote | 61.17 ± 1.30 | 60.29 ± 1.78 | 53.08 ± 3.10 | 63.71 ± 3.00 | 48.01 ± 0.84 | 54.13 ± 2.17 | 48.10 ± 0.43 | 52.61 ± 1.26 | **78.17 ± 4.14** |
| Contact | 90.43 ± 2.34 | 89.22 ± 0.66 | 94.14 ± 0.45 | 88.12 ± 1.50 | 74.19 ± 0.80 | 90.43 ± 0.17 | 89.91 ± 0.36 | 93.55 ± 0.52 | **100 ± 0.00** |

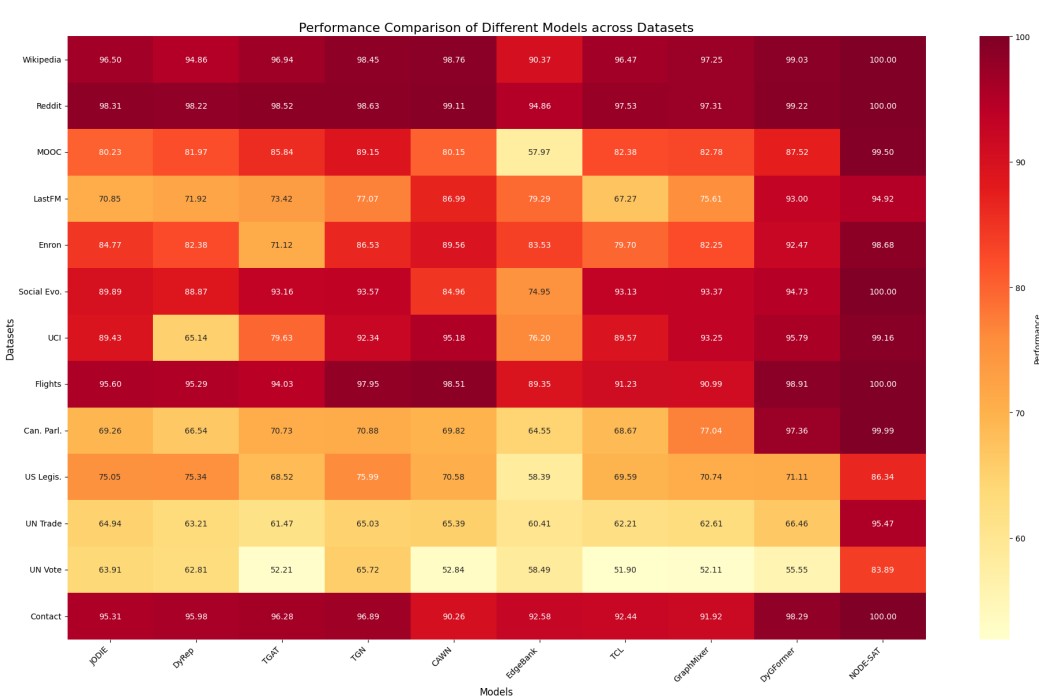

Figure 6: Heatmap of over-all performance(rnd)

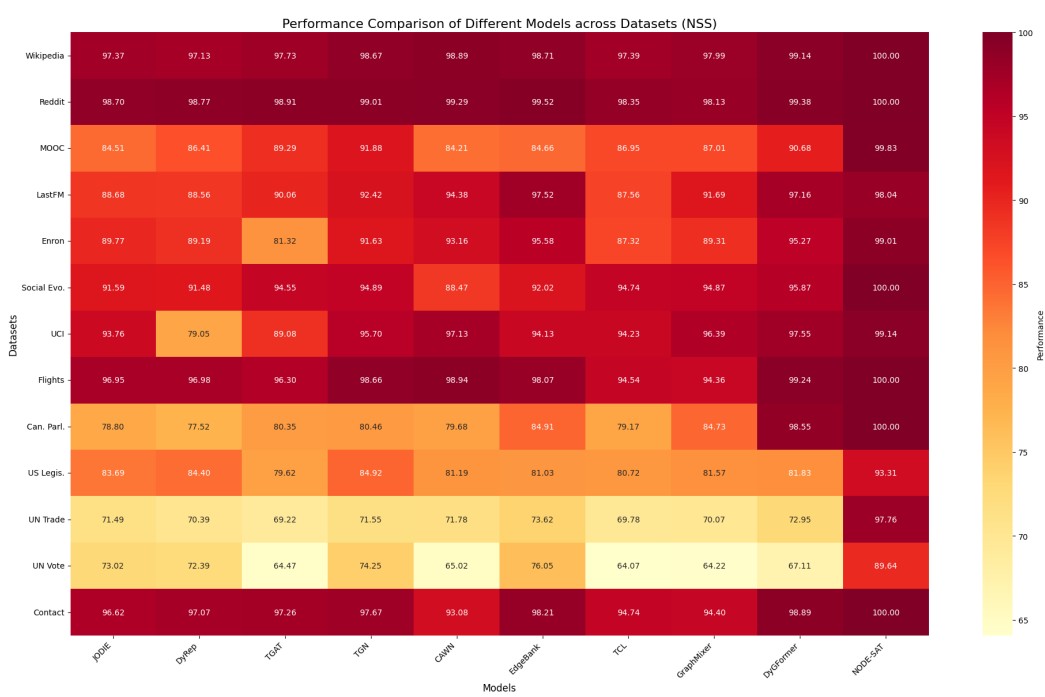

Figure 7: Heatmap of over-all performance(hist)

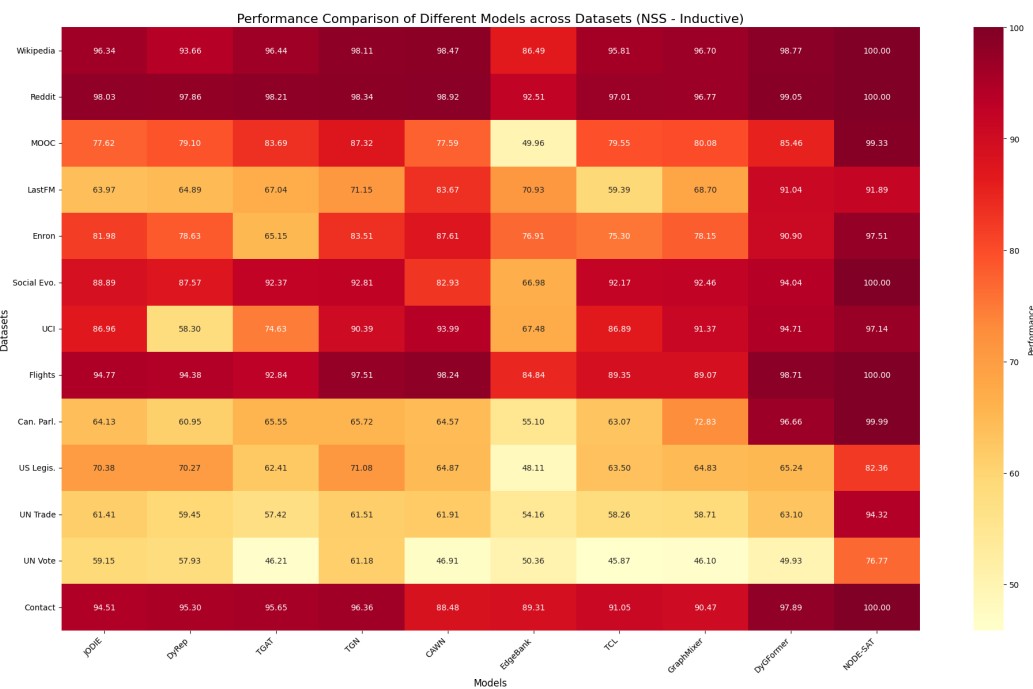

Figure 8: Heatmap of over-all performance(ind)

