# OpenReview forum: "NODE-SAT: Temporal Graph Learning with Neural ODE-Guided Self-Attention"
_ICLR.cc/2025/Conference — Submitted to ICLR 2025_

### Official Review · Reviewer_wuWu · 2024-10-18

**Soundness:** 2
**Presentation:** 1
**Contribution:** 2
**Rating:** 5
**Confidence:** 3

**Summary:**

The paper is about temporal graph learning, which is an interesting topic. The authors propose a novel temporal graph learning model that integrates Neural Ordinary Differential Equations (NODEs) with self-attention mechanisms. The paper is well written and well organized. However, there are several concerns in the current version of the paper that addressing them will increase the quality of this paper

**Strengths:**

1 Cutting-edge research directions.

2 Leading experimental results.

**Weaknesses:**

1 Authors should provide more research background in the abstract and introduction to help non-field readers better understand the paper.

2 The modules used by the authors in the model design seem to have been proposed by previous methods, but they are simply combined. Are there any new concepts or technologies proposed by the authors? At present, it is impossible to understand the challenges that the paper attempts to solve and the core contributions.

3 The authors should provide a more complete introduction to the dataset used, such as its size.

4 Through experiments, we can find that the method proposed in the paper has obvious improvements on some data sets. Perhaps the author has indeed done a commendable job, but the narrative is not good enough to capture the highlights.

5 The layout and structure of the entire paper could use considerable improvement, making it seem like a work in progress rather than a finished work.

**Questions:**

In the method section, did the author omit the introduction of the loss function?

---

> ### Author Response · Authors · 2024-11-27
> **Thanks for your careful review**
>
> We appreciate the reviewer's constructive feedback and thoughtful suggestions. Below, we address each of the concerns raised and provide clarifications and planned improvements.
>
> ## Lack of Research Background - W1:
> We acknowledge that the current abstract and introduction could be expanded to provide more context for non-expert readers. In the revised version, we will emphasize the growing significance of temporal graph learning across domains such as social networks, transportation, and e-commerce and explain the limitations of existing methods in capturing continuous-time dynamics and temporal dependencies.
>
> ## Clarity of Core Contributions - W2:
> We will highlight the technical innovations, such as the use of NODEs to model continuous-time temporal dynamics and the application of self-attention to capture interdependencies across time points. More clarification can be found in our discussion and reply to Reviewer xBHN.
> > Temporal graph data presents unique challenges.
>
> > How do the NODE and self-attention work together in NODE-SAT?
>
> ## Dataset Details - W3:
> The dataset details were attached in Appendix Section A.1 and we did a comprehensive description of each dataset.
>
> ## Narrative on Experimental Highlights - W4:
> We agree that the narrative could better emphasize the highlights of our experimental results. To address this, we will restructure the results section to explicitly showcase the significant improvements of NODE-SAT over state-of-the-art baselines via putting hist and ind negative sampling in supplementary.
>
> ## Paper Layout and Structure - W5:
> The current layout will be refined to improve readability and coherence:
> 1. Streamline transitions between sections for smoother flow.
> 2. Use the layout to separate the methodology part.
> 3. Ensure consistent formatting and presentation for figures, tables, and equations.
>
> ## Introduction of the Loss Function - Q1:
> We regret the omission of the loss function in the method section. As for the temporal link prediction task, we adopt the binary cross-entropy loss. We will explicitly describe the binary cross-entropy loss, including its mathematical formulation in the revised paper.
>
> ## Closing Statement
> We sincerely thank the reviewer wuWu for your valuable suggestions, which will significantly improve the quality and impact of our paper. We are committed to addressing these concerns thoroughly in the next revision and ensuring that the contributions and insights of NODE-SAT are clearly conveyed to the community. However, due to technical difficulties with accessing the local LaTeX format, we will update the revised paper at a later time.

---

> > ### Comment · Reviewer_wuWu · 2024-11-29
> >
> > Thanks to the authors for the reply, I'll consider keeping my score for now.

---

### Official Review · Reviewer_xBHN · 2024-11-04

**Soundness:** 3
**Presentation:** 2
**Contribution:** 3
**Rating:** 8
**Confidence:** 3

**Summary:**

The NODE-SAT model introduces a novel approach to temporal graph learning by combining Neural Ordinary Differential Equations  with self-attention mechanisms. Designed for temporal link prediction, NODE-SAT captures evolving temporal relationships between nodes by modeling continuous-time dynamics. NODE-SAT achieves strong performance across multiple benchmarks.

**Strengths:**

**Good contribution and engineering** : The use of NODEs for modeling the continuous evolution of node representations is, to the best of my knowledge, a novel approach **in the field of temporal graph learning**.

**Rigorous Experimental Results**: The experimental results are robust, and the authors follow a well-documented experimental pipeline that thoroughly assesses the performance improvements of NODE-SAT.

**Open-Source Code:** The release of the code is an excellent contribution, allowing for replication and further exploration of the experimental results. This transparency greatly enhances confidence in the impressive performance claims made by the authors.

**Weaknesses:**

**Some choice of presentation** , the big table is difficult to read i'll suggest to put hist and ind negative sampling in supplementary. Same remark for the graphics.

**Novelty** The core components introduced in NODE-SAT, particularly the use of Neural Ordinary Differential Equations (NODEs) for continuous-time modeling, rely on well-established methods in machine learning. While the combination of NODEs with self-attention is creative and effective for temporal graph learning, these elements themselves are not novel. Neural ODEs have been widely used for modeling continuous dynamics in various applications, and self-attention is a standard technique for capturing complex relationships in structured data. As such, the innovation here is more about the application of these established techniques rather than the introduction of fundamentally new methods.

**Questions:**

1. Impact of Multi-Hop Neighbors: What would be the effect on performance if the model used multi-hop neighbors instead of only 1-hop neighbors? Would incorporating multi-hop information further improve the model's ability to capture long-range temporal dependencies, or would it increase complexity without significant performance gains?

---

> ### Author Response · Authors · 2024-11-27
> **Thanks for your careful review**
>
> We appreciate the positive evaluation and thoughtful feedback provided by the reviewer. Below, we address the specific concerns and questions raised to further clarify our contributions and improve the paper.
>
> ## Presentation of Results - W1
> We acknowledge that the main table summarizing results across different negative sampling strategies (random, historical, inductive) may be overwhelming in its current form. In the revised version, we will:
> Move detailed results for historical (hist) and inductive (ind) negative sampling strategies to the supplementary material.
> Retain the summary table in the main paper, highlighting the most critical comparisons and key takeaways.
> Similarly, for graphics, we will condense the presentation by combining related plots into a single visualization where appropriate.
>
> ## Novelty of NODE-SAT - W2
> While we agree that Neural ODEs and self-attention mechanisms are established techniques, the novelty of NODE-SAT lies in its **synergistic application to temporal graph learning**.
>
> ### Temporal graph data presents unique challenges.
> #### 1. Continuous Temporal Dynamics: Temporal graphs capture interactions over time, requiring models to represent relationships that evolve continuously rather than in discrete snapshots. Many existing methods, such as RNN-based models, struggle to capture fine-grained temporal details, often aggregating data into fixed intervals, leading to a loss of temporal fidelity.
> #### 2. Complex Interdependencies: Temporal graphs exhibit intricate interdependencies between nodes and links, which may span multiple time scales. For instance, short-term interactions might influence long-term dynamics, requiring models to capture both.
>
> ### How do the NODE and Self-Attention work together in NODE-SAT?
>
> Neural Ordinary Differential Equations (NODEs) offer a natural way to model continuous-time dynamics:
> #### 1. Continuous-Time Modeling: NODEs allow us to model the hidden state evolution of nodes and links as a system of differential equations. This provides a seamless representation of continuous-time changes, avoiding the limitations of discrete aggregation methods.
> #### 2. Temporal Flexibility: NODEs can predict node and link states at arbitrary time points, enabling fine-grained control over temporal granularity. This is crucial for applications where temporal intervals between events vary significantly.
>
> Self-attention complements NODEs by addressing the challenges of capturing interdependencies:
> #### 1. Capturing Long-Range Dependencies: Temporal graphs often exhibit interactions that are non-local in both structure and time. Self-attention excels at modeling such dependencies by computing pairwise relationships between all nodes in a graph, irrespective of their distance.
> #### 2. Context-Aware Representations: Self-attention dynamically weighs the importance of different nodes and time points, enabling the model to focus on the most relevant interactions for each prediction task.
>
> Note that our ablation studies (Section 6.1) demonstrate the significant impact of this integration, particularly in improving stability and performance on challenging datasets like UN Trade and UN Vote. In the revised manuscript, we will explicitly emphasize these aspects of novelty to better communicate the unique contributions of NODE-SAT.
>
> ## Why We Focus on 1-Hop Neighbors - Q1
> We chose to limit the model to 1-hop neighbors to prioritize computational efficiency and avoid overfitting, especially in dense temporal graphs where multi-hop information can become overwhelming.
> Previous works, such as GraphMixer and DyGFormer, have shown that 1-hop neighborhood information is often sufficient for tasks like temporal link prediction. We leave the multi-hop with an efficient implementation as our future work exploration.

---

### Official Review · Reviewer_S6tg · 2024-11-04

**Soundness:** 2
**Presentation:** 3
**Contribution:** 2
**Rating:** 3
**Confidence:** 4

**Summary:**

This paper presents a temporal graph leanring algorithm that combine ODE with self attention mechanism.

**Strengths:**

1.  The paper is well written and easy to follow.

2. The results are pretty good.

**Weaknesses:**

1. The novelty of this work is not high, a mixture of self attention with neural ODE. It mainly uses some engineer effrot in my mind.

2. The results sounds too good to be true, In table 1, I saw many of the datasets can achieve 100% accuracy. Does mean that this paper almost close this area?

3. How is the complexity of this work.

**Questions:**

1. The novelty of this work is not high, a mixture of self attention with neural ODE. It mainly uses some engineer effrot in my mind.

2. The results sounds too good to be true, In table 1, I saw many of the datasets can achieve 100% accuracy. Does mean that this paper almost close this area?

3. How is the complexity of this work.

---

> ### Author Response · Authors · 2024-11-27
> **Thanks for your careful review**
>
> ## Addressing Concerns About Novelty - W1 and Q1
>
> We acknowledge the reviewer's observation regarding the novelty of combining Neural Ordinary Differential Equations (NODEs) and self-attention mechanisms. While both techniques have been studied independently, our key innovation lies in their synergistic integration within temporal graph learning, specifically tailored to the unique challenges of capturing continuous-time dynamics and temporal dependencies in evolving graphs. By introducing NODE-SAT, we demonstrate how this combination significantly enhances both model expressiveness and predictive power, outperforming existing state-of-the-art methods across a wide range of datasets.
>
> Importantly, as shown in Section 6.1 of the manuscript (Ablation Study), the incorporation of NODEs in our framework led to substantial performance improvements, particularly on challenging datasets like UN Trade (32.42% increase) and UN Vote (29.29% increase). These ablation study results validate the necessity and impact of our methodological contributions, which go beyond mere "engineering effort."
>
> ## Justifying Results That "Seem Too Good to Be True" - W2 and Q2
>
> The high performance of NODE-SAT, including instances of achieving 100% accuracy on certain datasets, is a reflection of its robustness and adaptability. These datasets (e.g., Wikipedia, Reddit, and Flights) exhibit well-defined temporal and structural patterns, which NODE-SAT is particularly adept at modeling due to its ability to capture fine-grained temporal dynamics and long-term dependencies. We emphasize that such results do not imply that the field is "closed," as the model's performance varies depending on the complexity of the dataset (e.g., UN Trade and US Legislature datasets).
>
> To ensure reproducibility and transparency, we have included detailed descriptions of our datasets, methodologies, and **code (available at the anonymous repository link in the paper)**. We also conducted extensive evaluations using multiple negative sampling strategies (random, historical, inductive) to validate our claims. Furthermore, we performed experiments with **multiple rounds with different random seeds**, reporting both the mean and variance (for some cases with 100&plusmn;0.00) to ensure the reliability of our results. The consistency of these results across diverse datasets also underscores the credibility of our experiments.
>
> ## Complexity Analysis - W3 and Q3
>
> We appreciate the reviewer's inquiry about the complexity of NODE-SAT. Our approach balances computational efficiency with model accuracy. Here is a detailed complexity analysis of NODE-SAT including the temporal link encoder, the neural ODE, and the self-attention module.
>
> Assume we have $N$ nodes, each containing an average of $K$ temporal links. The feature embedding dimension is $d$. The number of steps is $T$ for ODE.
>
> #### 1. The complexity of the temporal link Encoder is $O(NKd)$. Note that using the learnable encoding function $\phi(t) =cos(tw+b)$ adds constant overhead for $K$ links per node.
>
> #### 2. The complexity of the ODE solver is $O(NTKd^2)$. Specifically, $O(Kd^2)$ per ODE step for matrix multiplications for K temporal links.
>
> #### 3. The complexity for Self-attention is $O(N^2d)$.
> ###### (a) Query, Key, Value projection: $O(Nd^2)$
> ###### (b) Attention scores (dot product): $O(N^2d)$
> ###### (c) Softmax operation over $N$ nodes: $O(N^2)$
> ###### (d) Weighted sum for attention output: $O(N^2d)$
> For $N \gg d$, the overall self-attention complexity becomes $O(N^2d)$
>
> Therefore, the total complexity is $O(NKd)$ + $O(NTKd^2)$ + $O(N^2d)$.
> In practical implementations, $K$ (the number of temporal links) is typically small, and $T$ (ODE steps) is controlled for efficiency. The quadratic cost of self-attention can be mitigated by using sparse attention or efficient attention mechanisms.
>
> ## Closing Statement
> We thank the reviewer for their valuable feedback and for raising these pertinent questions. We hope our clarifications effectively address the concerns, demonstrating the novelty, soundness, and reproducibility of NODE-SAT. We are committed to advancing the field and welcome further dialogue to refine our contributions.

---

### Meta-Review · Area_Chair_TGS7 · 2024-12-20

**Metareview:**

This paper explores temporal graph learning by integrating Neural ODEs with self-attention mechanisms. While the topic is relevant and the results show promise, the paper lacks sufficient novelty, as it primarily combines existing methods without introducing significant new concepts. Concerns about the saturated results (e.g., 100% accuracy on several datasets) and unclear scalability further weaken the submission. Reviewers also noted issues with clarity, incomplete methodology descriptions, and insufficiently detailed dataset information. While the authors provided detailed responses, the revisions did not address the core concerns. Overall, the paper requires substantial improvements to reach publication standards. For example, the authors are encouraged to use modern large-scale datasets like temporal graph benchmark (TGB) as traditional datasets are saturated.

**Additional Comments On Reviewer Discussion:**

During the rebuttal, reviewers raised concerns about the paper’s novelty, credibility of results, scalability, and presentation clarity. R1 and R3 questioned the originality of combining NODEs with self-attention, which the authors defended as innovative in temporal graph learning. R1 and R2 flagged 100% accuracy on datasets as overfitting, which authors explained via dataset characteristics and reproducibility measures. R3 highlighted missing details (e.g., loss function, dataset descriptions), which authors addressed partially. While the responses clarified some issues, they did not resolve concerns about limited novelty and unclear generalizability. These factors weighed heavily in the final decision to reject the paper.

---

### Decision · Program_Chairs · 2025-01-22

Reject